# Opposing Effects of Prior Infection versus Prior Vaccination on Vaccine Immunogenicity against Influenza A(H3N2) Viruses

**DOI:** 10.3390/v14030470

**Published:** 2022-02-25

**Authors:** Annette Fox, Louise Carolan, Vivian Leung, Hoang Vu Mai Phuong, Arseniy Khvorov, Maria Auladell, Yeu-Yang Tseng, Pham Quang Thai, Ian Barr, Kanta Subbarao, Le Thi Quynh Mai, H. Rogier van Doorn, Sheena G. Sullivan

**Affiliations:** 1WHO Collaborating Centre for Reference and Research on Influenza, Royal Melbourne Hospital, Peter Doherty Institute for Infection and Immunity, Melbourne, VIC 3000, Australia; louise.carolan@influenzacentre.org (L.C.); vivian.leung@mh.org.au (V.L.); ian.barr@influenzacentre.org (I.B.); kanta.subbarao@influenzacentre.org (K.S.); sheena.sullivan@influenzacentre.org (S.G.S.); 2Department of Infectious Diseases, University of Melbourne, Peter Doherty Institute for Infection and Immunity, Melbourne, VIC 3000, Australia; sen.khvorov@unimelb.edu.au (A.K.); ryan.tseng@influenzacentre.org (Y.-Y.T.); 3National Institute of Hygiene and Epidemiology, Ha Noi 100000, Vietnam; hvmp@nihe.org.vn (H.V.M.P.); pqt@nihe.org.vn (P.Q.T.); lom9@hotmail.com (L.T.Q.M.); 4Department of Microbiology and Immunology, Peter Doherty Institute for Infection and Immunity, University of Melbourne, Melbourne, VIC 3000, Australia; mauladell@chdr.nl; 5Oxford University Clinical Research Unit, Wellcome Africa Asia Programme, National Hospital of Tropical Diseases, Ha Noi 100000, Vietnam; rvandoorn@oucru.org; 6Centre of Tropical Medicine and Global Health, Nuffield Department of Clinical Medicine, University of Oxford, Oxford OX3 7LG, UK

**Keywords:** influenza, vaccination, infection, immunogenicity, antibodies, pre-existing immunity, memory

## Abstract

Prior vaccination can alternately enhance or attenuate influenza vaccine immunogenicity and effectiveness. Analogously, we found that vaccine immunogenicity was enhanced by prior A(H3N2) virus infection among participants of the Ha Nam Cohort, Viet Nam, but was attenuated by prior vaccination among Australian Health Care Workers (HCWs) vaccinated in the same year. Here, we combined these studies to directly compare antibody titers against 35 A(H3N2) viruses spanning 1968–2018. Participants received licensed inactivated vaccines containing A/HongKong/4801/2014 (H3N2). The analysis was limited to participants aged 18–65 Y, and compared those exposed to A(H3N2) viruses circulating since 2009 by infection (Ha Nam) or vaccination (HCWs) to a reference group who had no recent A(H3N2) infection or vaccination (Ha Nam). Antibody responses were compared by fitting titer/titer-rise landscapes across strains, and by estimating titer ratios to the reference group of 2009–2018 viruses. Pre-vaccination, titers were lowest against 2009–2014 viruses among the reference (no recent exposure) group. Post-vaccination, titers were, on average, two-fold higher among participants with prior infection and two-fold lower among participants with 3–5 prior vaccinations compared to the reference group. Titer rise was negligible among participants with 3–5 prior vaccinations, poor among participants with 1–2 prior vaccinations, and equivalent or better among those with prior infection compared to the reference group. The enhancing effect of prior infection versus the incrementally attenuating effect of prior vaccinations suggests that these exposures may alternately promote and constrain the generation of memory that can be recalled by a new vaccine strain.

## 1. Introduction

Influenza viruses can evolve relatively rapidly because viral RNA replicates without proofreading. Substitutions that increase virus fitness or facilitate escape from host immune responses are positively selected. Influenza virus hemagglutinin (HA) mediates infection and accumulates mutations faster than other influenza virus proteins due to selection pressure from HA-reactive antibodies that block infection [1,2,3]. A key consequence of HA antigenic evolution, termed antigenic drift, is that influenza viruses re-infect people throughout their lives. Therefore, influenza vaccines are frequently re-formulated to match circulating strains, and re-administered [4]. Of note, influenza A(H3N2) viruses have undergone more antigenic change than A(H1N1) viruses [5]. Accordingly, recent studies estimate that re-infection occurs more frequently for A(H3N2) compared to A(H1N1) viruses [6].

The effect of adaptive immune memory on responses to variant viruses has been debated since the 1950’s, when Davenport and others showed that exposure to new influenza virus strains induced higher antibody titers against priming strains that were typically encountered early in life [7,8,9]. From these studies, Francis formulated the “original antigenic sin” hypothesis, which postulates that responses against minimal epitopes that are preserved from past strains are preferentially recalled (back-boosted) at the expense of generating responses against epitopes that are unique to the new strain [10]. This was considered “sinful” because the cross-reacting antibodies induced had relatively poor neutralizing titers against the new strain [10,11]. Contemporary studies have since demonstrated that the extent of antibody back-boosting induced by new A(H3N2) strains diminishes with increasing temporal and antigenic distance from the new strain, even though back-boosting can extend to strains encountered early in life [12,13,14].

In the 1970s, Hoskins et al. observed that boys vaccinated for the first time had lower A(H3N2) virus attack rates than boys who had also been vaccinated in the prior year(s) [15,16], raising concern that pre-existing immunity may attenuate vaccine-induced protection against influenza illness. However, Hoskins et al. also observed that A(H3N2) attack rates were lower among boys who had prior A(H3N2) infection, and concluded that infection induced greater immunity than vaccination [15,16]. Subsequent studies have confirmed that repeated annual administration of influenza vaccine can be associated with reduced vaccine effectiveness (VE) [17,18,19,20], and reduced antibody titer and titer rises [21,22,23,24,25,26]. Repeated vaccination has mainly been associated with the attenuation of VE and immunogenicity against A(H3N2) viruses rather than against A(H1N1) and B viruses [27]. This agrees with consistent reports, across years and geographic regions of poor VE (<40%) against influenza A(H3N2) compared to A(H1N1) and B viruses [19,27,28,29,30,31,32,33,34,35,36]. The exception being that estimated VE against A(H3N2) may be higher in young children [34,37], who will have less pre-existing immunity. These phenomena indicate that pre-existing immunity may limit the capacity for vaccination to update immunity against new A(H3N2) strains. 

Effects of prior vaccination on VE have varied between studies [27] and seasons [38,39]. The antigenic distance hypothesis, supported by mathematical modeling, predicts that a prior vaccine will negatively interfere with a current vaccine when the antigenic distance between successive vaccine strains is small, and that this will attenuate protection when the antigenic distance between the vaccine and subsequent epidemic strains is large [38,40]. The antibody focusing hypothesis suggests that recalled memory B cells competitively dominate and focus responses on epitopes shared with previous strains, which could attenuate protection if epidemic strains acquire mutations within epitopes upon which antibodies are focused [41]. Monto et al. have suggested that the term “negative antigenic interaction” best captures the immune mechanism underlying repeated vaccination effects, and stressed the need to identify mechanisms that account for negative effects of prior vaccination versus potentially positive protective effects of prior infection [42].

In 2016, we conducted a study among 100 vaccine-naïve adults who had participated in influenza surveillance for nine years as part of the Ha Nam community household cohort in Viet Nam [14]. Recent A(H3N2) virus infection was associated with substantially and significantly reduced detection of symptomatic A(H3N2) infection in the season after vaccination [14]. This corresponded to greater antibody responses against A(H3N2) viruses among vaccinees who had recent infection [14]. In the same year, and using the same vaccine formulation, we also conducted a study among 157 Australian healthcare workers (HCWs), with varying prior exposures to vaccination, but unknown prior exposures to infection [43]. Vaccine-naïve HCWs and those who had been vaccinated only once before demonstrated greater post-vaccination responses against the A(H3N2) component of the vaccine than frequently vaccinated (3+ prior vaccines) HCWs [43]. Here, we have combined these studies to compare titers across 35 A(H3N2) antigens spanning 1968 to 2018 among participants with prior exposure through infection versus vaccination.

## 2. Materials and Methods

### 2.1. Vaccination Study Designs

The Ha Nam Cohort and the vaccination sub-study have been described previously [14,44]. In brief, 945 members of 270 households commenced participation in active surveillance for influenza infection causing RT-PCR-confirmed illness or seroconversion without illness in December 2007. In November 2016, 100 participants aged at least 18 Y, who had participated in all investigations to detect laboratory confirmed influenza infection received Southern Hemisphere trivalent inactivated split egg-grown influenza vaccine (Vaxigrip, Sanofi). This included 28 who lacked A(H3N2) virus infection since 2007, and 72 of similar sex and age who had at least one A(H3N2) virus infection. Venous blood was collected before and at multiple post vaccination time points, including days 21 and 280–282. The study was approved by ethics committees of the University of Melbourne (1646470), the National Institute of Hygiene and Epidemiology in Viet Nam (IRB-VN01057 – 08/2016), and the Oxford Tropical Medicine Research Ethics Committee (30–16).

The HCW vaccine study has been described previously [43]. In brief, HCWs aged 18-65 years were recruited in April 2016 when attending the staff influenza vaccination clinic of the Royal Melbourne Hospital, Victoria, Australia to receive the 2016 Southern Hemisphere quadrivalent inactivated split egg-grown influenza vaccine (Fluarix Tetra 2016, Glaxo Smith Kline). Participants were asked if they had been vaccinated against influenza in the preceding 5 years since 2011. Venous blood was collected prior to vaccination and post vaccination on days 21–28 and 210–227. The study was approved by the human research ethics committee at the Royal Melbourne Hospital (reference number: HREC/16/MH/11).

### 2.2. Definition of Prior Exposure Groups, and Criteria for Inclusion in Each Group

Four groups were defined based on prior exposure by infection or vaccination to A(H3N2) viruses going back to A/Perth/16/2009-like viruses included in the 2011 vaccine (Table 1). These groups were referred to as “no recent exposure”, “recent infection”, “1–2 prior vaccinations” and “3–5 prior vaccinations”. Ha Nam cohort participants were all vaccine-naïve, so were assigned to the “no recent exposure” group if A(H3N2) virus infection was not detected since December 2007, or to the “prior infection” group if infection was detected since 2009. HCWs were assigned to prior vaccination groups based on the number of times they were vaccinated in the preceding 5 years (HCWs) since 2011. HCWs who had no vaccinations in the preceding 5 years were excluded because it was not known if they had recent infection.

Inclusion and exclusion criteria for this analysis are shown in Figure 1. Participants who developed ILI with A(H3N2) infection confirmed by RT-PCR in the season after vaccination were excluded from both comparison groups because infection was expected a priori to lead to higher post-season titers, hindering interpretation. Ha Nam vaccinees were excluded if aged > 65 Y. Further, we excluded two participants who were last infected in 2008 with an A/Brisbane/10/2007-like (H3N2) virus in order to match the range of strains that HCWs were exposed to by vaccination between 2011 and 2015. HCWs were excluded if they did not complete the study, if vaccination histories were incomplete, or if insufficient sera remained for titration against 35 viruses. HCWs who had less than five prior vaccinations were under-represented, and all were selected. A similar number of HCWs with five prior vaccinations were selected randomly after stratifying by sex in order to obtain similar sex ratios to the other exposure groups.

### 2.3. Viruses

Viruses that circulated between 1968 and 2018 (*n* = 35) were propagated in mammalian cell lines and/or in eggs for use in serology. Where necessary, viruses were plaque-selected on Madin Darby Canine Kidney cells that had been transfected with human 2,6-sialtransferase (MDCK-SIAT cells) to produce stocks that lacked NA T148X or D151X substitutions, which can reduce the sensitivity of the assay for detecting HI antibodies [45]. Virus HA genes were sequenced and aligned with vaccine strains and strains circulating in the Ha Nam Cohort during the study period (Appendix A). A(H3N2) strains included in vaccines between 2011–2016 (Table 1) were further scrutinized for substitutions at 131 amino acid positions that have been associated with antigenic variation and assigned to antigenic sites [46] (Appendix A).

### 2.4. Serology

Sera from both studies were tested in hemagglutination inhibition (HI) assay against 35 A(H3N2) viruses (Table 1). Viruses were selected to represent each main antigenic or genetic cluster detected between 1968, when A(H3N2) emerged in humans, and 2018, four years after the strain included in the 2016 vaccine (A/Hong Kong/4801/2014). HI assays were performed according to WHO Global Influenza Surveillance Network protocols [47], with the exception that the red blood cell percentage and volume per well were 1% and 25 µL, respectively. Quality controls were performed for each new batch of red blood cells used to enable comparison of titers across multiple viruses and time points. HI titer reading was automated using a CypherOne reader (InDevR, Inc., Boulder, CO 80301).

### 2.5. Analysis

To compare antibody responses across strains, and between prior exposure groups, titers were presented as fitted landscapes across strains, and as averages for sets of strains, focusing on strains circulating since 2009. Generalized additive models (GAMs) were used to fit log_2_ titers and log_2_ titer differences (geometric ratios) between timepoints against 35 A(H3N2) viruses arranged temporally. Fitted titer estimates were presented with 95% confidence intervals. An antibody landscape package (https://github.com/acorg/ablandscapes, accessed on 22 August 2021), which uses Lowess models, was used to fit log_2_ titers against viruses arranged on a two-dimensional map of antigenic distances, and present estimated titers as contours [13]. Strains that circulated prior to a participants’ birth year were excluded to reduce the impact of age on landscapes. Birth years ranged from 1950 to 1996, meaning that up to 10 strains circulating between 1968 to 1995 could have been excluded. We used the GAM function from the R package mgcv and accounted for repeated measurements on each individual through specification of a random effect [48]. To further compare effects of prior vaccination versus prior infection, the no prior exposure group was assigned as a reference, and the titer ratios were estimated for each prior exposure group compared to the reference. Ratios were estimated using the linear model function (titre_ratio~lm(log(titre)~exposure_group) of the tidyverse package in R. All plots were generated with the ggplot2 package in R [49].

## 3. Results

### 3.1. Characteristics of Participants in Each Prior A(H3N2) Exposure Group

Eighty-six of 100 Ha Nam vaccinees, and 41 of 112 eligible HCW vaccinees were included in this analysis (Figure 1). HCWs included 13/13 with 1–2 prior vaccinations; 8/8 with 3–4 prior vaccinations; and 20/91 with 5 prior vaccinations. These participants were classified according to exposure to A(H3N2) viruses by infection or vaccination since 2009 (Figure 1, Table 2). Sex and age distributions, as well as sampling times, were similar across exposure groups (Table 2).

### 3.2. Vaccine-Induced Antibody Responses against A(H3N2) Viruses, Comparing Participants with Prior Infection versus Vaccination to Those Lacking Recent Exposure

Compared to participants with no recent exposure, participants with prior vaccination had higher pre-vaccination antibody titers against 2009–2014 strains, approximating strains present in prior vaccines (Figure 2A,B, Appendix A). Pre-vaccination titers were higher against a greater range of strains, extending back to circa 2002, among participants with prior infection (Figure 2A). Titers against strains circulating before 2000 were not consistently different between prior exposure groups (Figure 2A).

Post vaccination, a clear hierarchy emerged, with titers against strains spanning 2009–2018 being higher among participants with prior infection, and lower among those with 3–5 prior vaccinations compared to participants with no recent exposure (Figure 2B,C). Participants with 1–2 prior vaccinations had intermediate post-vaccination antibody titers against recent strains, which did not exceed the titers of participants with no recent exposure. These trends reflected titer rise, which was negligible among participants with 3–5 prior vaccinations and relatively poor among participants with 1–2 prior vaccinations compared to those with no recent exposure (Figure 2D). 

Post-season titers remained relatively high among participants with prior infection, but not among those with prior vaccination compared to the group with no recent exposure (Figure 2E). This reflected better maintenance of titer rise from pre-vaccination levels among participants with prior infection, but poorer maintenance among participants with prior vaccination, compared to those with no recent exposure (Figure 2F). HCWs who developed ILI in the season after vaccination were excluded from these comparisons. However, separate analysis of this group showed that infection was substantially more immunogenic than vaccination, inducing titer rises exceeding four-fold and extending across a broad range of strains (Appendix A). The average titer rise against cell-grown viruses was notably poor post vaccination, but exceeded four-fold post season (after infection) (Appendix A), consistent with the antigenic differences between egg-grown vaccines and circulating strains and their cell-grown equivalents.

The impact of prior infection versus prior vaccination on vaccine responses was further examined by assigning the no recent exposure group as a reference and estimating titer ratios to reference for each prior exposure group. Pre-vaccination geometric mean titers (GMTs), shown for all groups, were generally lower among the no recent exposure group (Figure 3A). Accordingly, pre-vaccination titer ratios to reference exceeded one for all viruses for the prior infection group, and for a narrower range of viruses for the prior vaccination groups (Figure 3B). Post vaccination, titer ratios for most viruses were similar to pre-vaccination levels for the prior infection group, but had decreased for the prior vaccination groups, falling substantially below one for participants with 3–5 prior vaccinations (Figure 3C). Titer ratios for A/Victoria/361/2011 (Vi11) were higher among groups with prior exposure at all time points, indicating that antibodies may be particularly well maintained and back-boosted against this virus. Post season, the prior infection group maintained high ratios, the ratios for the 3–5 prior vaccination group were still largely below 1, and the 1–2 prior vaccination group appeared to maintain titers no worse or better than the no recent exposure reference group (Figure 3D). 

The combined data indicate that the immunogenicity of the 2016 Southern Hemisphere vaccine against A(H3N2) viruses was enhanced by recent infection but attenuated by annual administration (3+ years) of vaccines containing strains similar to those causing infection.

### 3.3. Investigation of Antigenic Site Substitutions across Vaccine Strains and the Potential for Antibody Focusing

The antibody focusing theory suggests that memory B cells against epitopes that are retained from past strains are recalled, and competitively dominate vaccine responses so that memory is not induced against variant epitopes [41]. The number of antigenic site positions of HA that differed between each prior vaccine strain from 2011 to 2015 and the 2016 vaccine strain, A/Hong Kong/4801/2014 (HK14e) ranged from 10 for A/Victoria/361/2011 (Vi11e) to 14 for A/Perth/16/2009 (Pe09e) (Table 3, Appendix A). By comparison, 20 antigenic site positions were substituted across vaccine strains used between 2011 and 2016 (Table 3). Hence, the pool of vaccine-reactive memory may decline with each new vaccine strain encountered if memory is not induced against antigenic sites bearing substitutions. We investigated whether successive exposure to different vaccine strains affected antibody titer and titer rise distribution across a two-dimensional map of virus antigenic distances (Figure 4). Titer/titer rise distribution differed between HCWs who had one versus two prior vaccinations, indicating that the pool of A/Hong Kong/4801/2014 (HK14) vaccine-reactive memory may have been altered by exposure to A/Texas/50/12 (Tx12) prior to A/Switzerland/9715293/2013 (Sw13) vaccine. Titer distribution differed again among HCWs who had five prior vaccinations.

## 4. Discussion

The analysis presented here shows that the immunogenicity of inactivated split egg-grown influenza vaccine against A(H3N2) viruses was enhanced by recent A(H3N2) virus infection but attenuated by recent vaccination. These findings are reminiscent of Hoskins’ studies, as described in the introduction [16], and with animal models, which indicate that infection provides more potent priming than vaccination [50,51,52]. Additionally, a recent study from Japan found that the negative effect of prior vaccination on VE can be mitigated by prior infection [53]. Most repeat vaccination studies, including the current study, involve inactivated egg-grown virus vaccines. Inactivated cell-grown, recombinant HA, and adjuvanted vaccines have been developed in recent years, and early evaluations indicate that they are more immunogenic [54,55,56,57,58] and effective [59,60,61] than standard egg-grown virus vaccines. It will be important to determine whether these vaccine formulations can overcome or alleviate the attenuating effects of repeated vaccination. While vaccine effectiveness against A(H3N2) viruses is poor and attenuated by prior vaccination, current estimates suggest that vaccination in the current and prior season affords better protection against A(H3N2) than being vaccinated in the prior season only [62]. This suggests that it is better to vaccinate annually than in alternate years. However, there have been no formal comparisons of VE among people vaccinated in alternate versus successive years to determine whether there is any benefit to protection in the vaccinated years, and whether this outweighs the increased risk of infection in unvaccinated years. Additionally, the results presented here indicate that vaccine immunogenicity was particularly poor in people who had been vaccinated in three to five, as opposed to one to two, prior years, indicating a need to determine the impact of multiple years of vaccination on VE. 

The findings presented here suggest that existing memory (B cell) responses induced by infection enhance vaccine immunogenicity, and in turn, that repeatedly vaccinated individuals lack memory responses that cross-react with the vaccine antigen. Several studies have indicated that, similar to antibody responses, B cell responses decline with repeated annual vaccination [21,22,63]. Some suggest that existing antibodies may sequester (mask or clear) antigens and thereby dampen adaptive immune cell activation [40,63]. However, poor vaccination responses were not associated with high pre-vaccination antibody titers in the present study. Recent clinical trials demonstrate that inactivated influenza vaccines are immunogenic and effective in previously-naïve infants [64,65,66,67], indicating that these vaccines can prime naïve B cells. However, memory B cells are intrinsically programed to out-compete naïve B cells upon B cell receptor engagement [68,69,70], and it has been proposed that this could account for the observation that antibody responses to A(H1N1pdm09) vaccines were highly focused on epitopes shared with previously encountered A(H1N1) viruses in selected age groups [41,71,72,73]. Similarly, it was speculated that vaccination in the 2018/19 season was less effective against clade 3c3a A(H3N2) viruses in persons born before 1983 because HA S159T/Y substitutions emerged in 1983 and have been retained in subsequent strains, except those belonging to clade 3c3a [32,35]. It is plausible that responses diminish with repeated vaccination due to antigenic change across vaccine strains, in combination with competitive memory dominance. While memory may become limited against variant-neutralizing antibody epitopes with repeated vaccination, memory against more conserved non-neutralizing epitopes could be a source of substantial competition [74,75].

Direct comparisons of immune responses induced by infection and vaccination indicate that infection is more immunogenic. Specifically, A(H3N2) infection has been shown to induce more cross-reactive and less clonally expanded anti-hemagglutinin antibodies than influenza vaccination [75,76]. Infection also induces higher frequencies of CD4+ T cells with increased functional capacity than are induced by inactivated vaccine [77]. While the broader back-boosting capacity of infection has been interpreted to indicate that infection is a more potent driver of original antigenic sin than vaccination [75,78], this does not translate to antibody responses. It has been reported that infection promotes the maturation of memory B cell affinity against the HA head of infecting A(H3N2) strains [78]. Adjuvants increase the capacity of inactivated vaccine to induce naïve B cells and reduce the effects of prior vaccinations [79]. It is highly plausible that infection will induce greater naïve B cell differentiation than vaccination with inactivated virus because of greater innate immune stimulation [80] and antigen retention so that naïve B cells can be engaged after the memory B cell response starts to contract [81].

This study had several limitations. Sample sizes were small, particularly for HCWs who had less than five prior vaccinations. The effects of prior infection versus vaccination reported here could be biased by ethnic and socioeconomic differences between community members from Viet Nam and HCWs from Australia. However, we could detect effects of vaccination history within the HCW cohort, consistent with our previous studies [25], and with studies showing effects on VE in various populations and countries [17,18,53]. HCW infection histories were unknown, so we were unable to account for effects of prior infections, which may have alleviated the attenuating effects of prior vaccination [53]. These limitations could be remedied by the longitudinal study of other populations who have more varied uptake of the influenza vaccine, and who are monitored for clinical and sub-clinical influenza infection. 

## 5. Conclusions

In conclusion, the results presented here indicate that recall of existing memory can enhance antibody titers induced by inactivated egg-based influenza vaccines, but may concurrently limit the generation of memory against variant epitopes of vaccines, accounting for the detrimental effect of repeated vaccination. Direct examination of how exposure history affects the magnitude and repertoire of memory B cells induced by different vaccine formulations will be important to inform the development of vaccines that provide better protection against A(H3N2) viruses.

## Figures and Tables

**Figure 1 viruses-14-00470-f001:**
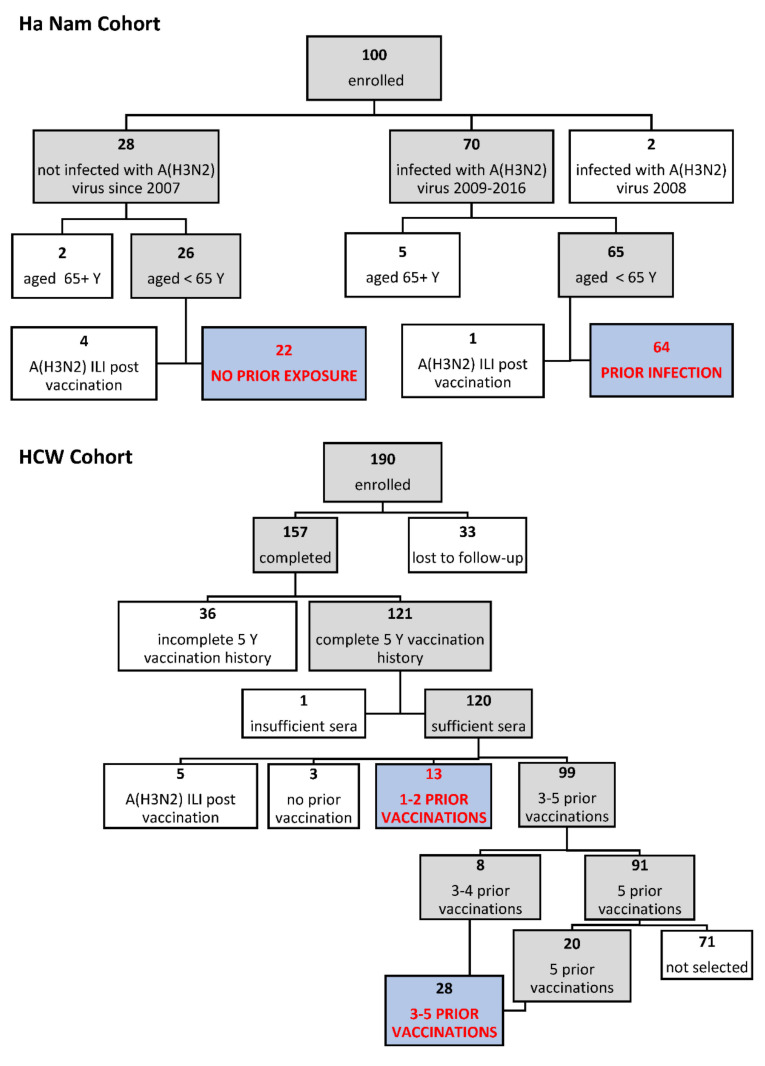
Participant inclusion and exclusion criteria. Shaded boxes indicate participants that met each of the inclusion criteria considered; clear boxes indicate those that were excluded. Blue shaded boxes indicate the four groups analyzed in this study.

**Figure 2 viruses-14-00470-f002:**
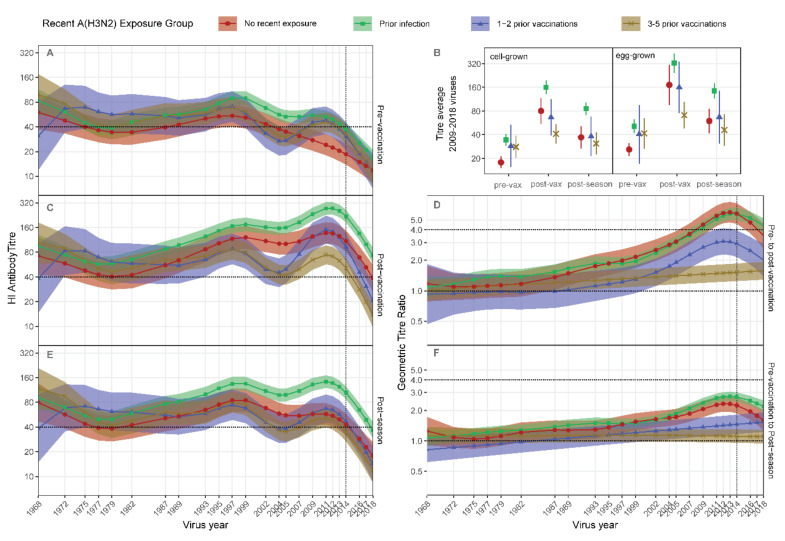
Antibody landscapes induced by vaccination differ between groups defined by prior exposure to A(H3N2). Participants were grouped according to recent prior exposure to A(H3N2) by infection or vaccination (legend), then Generalized Additive Models (GAMs) were used to fit HI titers across 35 viruses. (**A**), Pre-vaccination titer model, (**C**), Post vaccination titer model, (**D**), Pre- to Post-vaccination titer ratio model, (**E**), Post-season titer model, (**F**), Pre-vaccination to Post-season titer ration model. Shading indicates 95% confidence intervals for the models. Numbers/group are presented in Table 2. Dashed vertical lines indicate the vaccine strain. Dashed horizontal lines indicate sero-positive or sero-conversion thresholds. (**B**), titers averaged against 2009 to 2018 viruses that were cell-grown or egg-grown are presented as GMTs for each time-point and prior exposure group. Error bars represent 95% confidence intervals.

**Figure 3 viruses-14-00470-f003:**
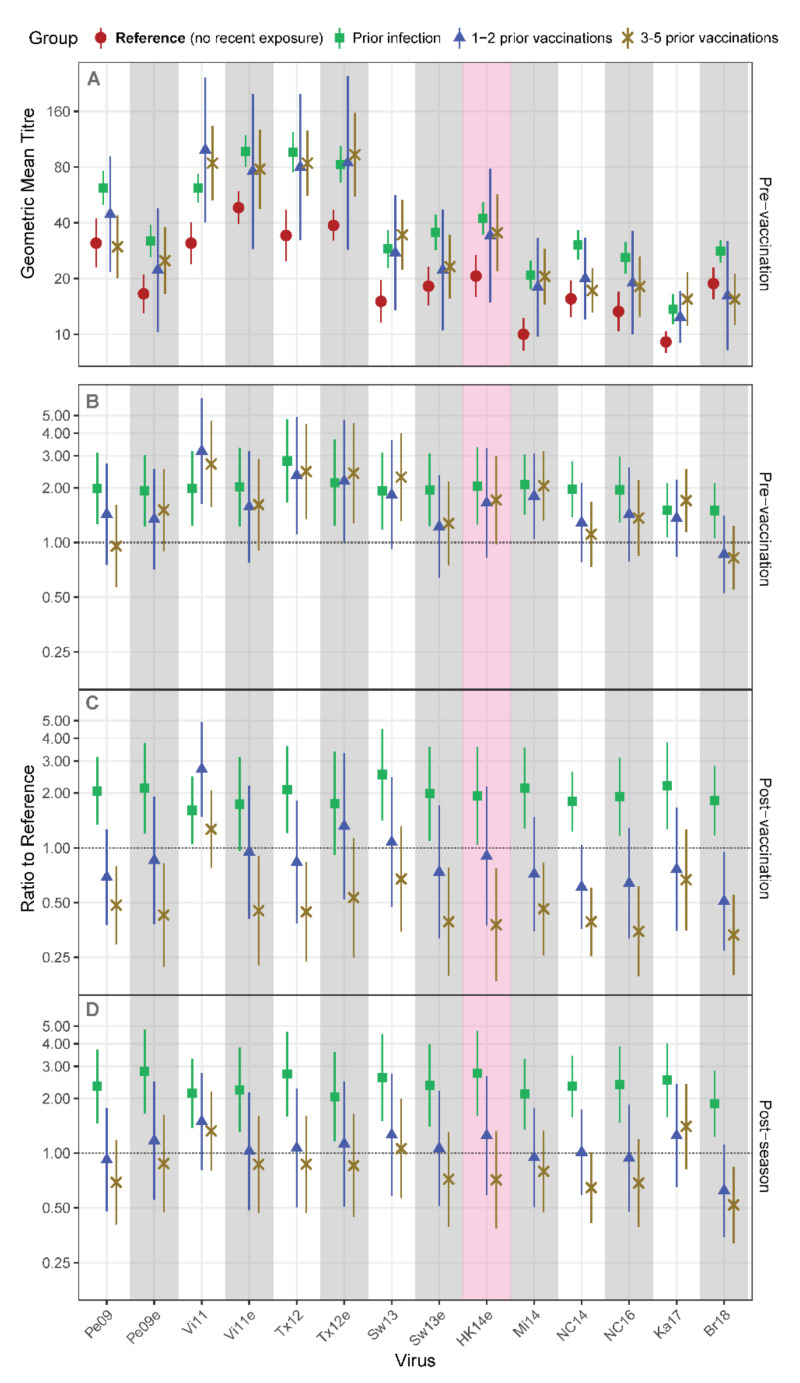
Recent strain antibody titers of groups with documented prior exposure to A(H3N2) compared to a reference group with no recent exposure. Ha Nam Cohort participants who lacked recent A(H3N2) exposure were assigned as the reference group. Pre-vaccination titers against A(H3N2) viruses are shown for each prior exposure group in (**A)**, and are presented as ratios compared to the reference group in (**B**). Post vaccination and Post season ratios to reference group are presented in (**C**) and (**D**). Numbers per group are given in Table 3. Results are presented as geometric mean titers (**A**) or as estimated mean ratios to reference (**B**–**D**) with 95% confidence intervals. The red shaded panel indicates the current vaccine strain.

**Figure 4 viruses-14-00470-f004:**
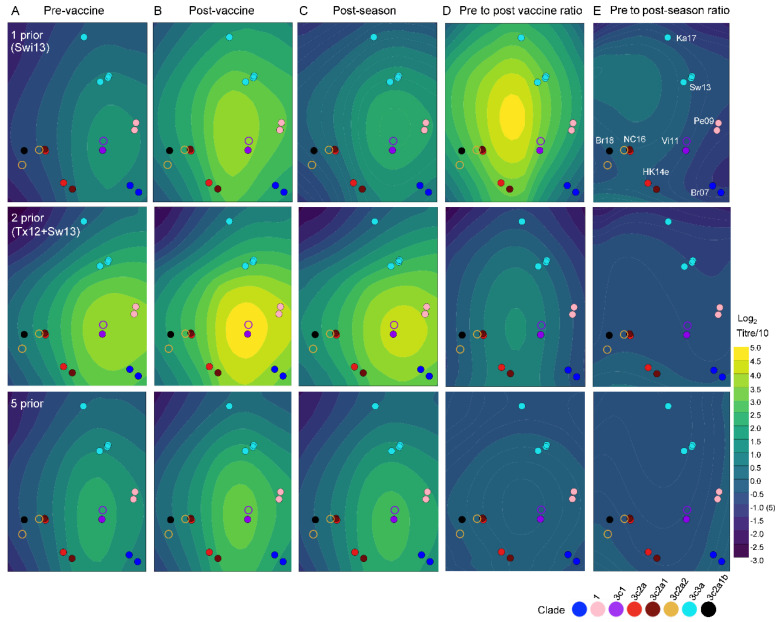
The strain coverage of antibodies induced by vaccination varies with vaccination history. Estimated antibody titer and titer rise landscapes are presented as contours over a two-dimensional map of A(H3N2) virus antigenic distances. Participants are grouped by number of prior vaccinations (rows). Panels show fitted titers Pre-Vaccination (**A**), Post-Vaccination (**B**), or Post-Season (**C**), and titer ratios Post-vaccination (**D**) or Post-Season (**E**). Each circle represents a virus on the map, colored by (sub)clade, assigned since 2009. Abbreviated virus names are shown in panel (**E**). Solid circles indicate viruses against which participant sera were titrated; other viruses are indicated by open circles. Model estimates were generated from 8 HCWs with 1 prior vaccination, 5 with 2 prior vaccinations, and 20 with 5 prior vaccinations.

**Table 1 viruses-14-00470-t001:** Viruses used for antibody assays.

Year	Virus Designation ^a^	Abbreviation	Passage ^b^	Vaccine Year ^c^
1968	A/Bilthoven/16190/68	Bi68	X, MDCK3	-
1972	A/Bilthoven/21793/72	Bi72	MDCK3	-
1975	A/Bilthoven/1761/76	Bi76	MDCK3	-
1977	A/Bilthoven/2271/76	Bi76b	X, MDCK3	-
1979	A/Netherlands/233/82	Ne82	tMK1, MDCK4	-
1982	A/Philippines/2/82	Ph82	MDCKX, 2	-
1987	A/Netherlands/620/89	Ne89	X, tMK1, MDCK3	-
1989	A/Netherlands/823/92	Ne92	X, MDCK3	-
1993	A/Netherlands/179/93	Ne93	X, MDCK3	-
1995	A/Netherlands/178/95	Ne95	293T, MDCK4	-
1997	A/Tasmania/1/97	Ta97	MDCK7	-
1999	A/Netherlands/301/99	Ne99	MDCK5	-
1999	A/Townsville/2/99	Tv99	MDCK2, SIAT1 ^p^	-
2002	A/Philippines/472/02	Ph99	MDCK6 ^p^	-
2002	A/Fujian/411/02	Fu99	X, MDCK9, SIAT1	-
2004	A/Victoria/511/04	Vi04	MDCKx, 2 ^p^	-
2004	A/New York/55/04e	NY04e	SPFCK3, Egg6	2006
2005	A/Thailand/409/05	Th05	P2, MDCK2 ^p^	-
2005	A/Wisconsin/67/05e	Wi05e	SPFCK3, Egg8	2007
2007	A/Brisbane/10/07	Br07	MDCKX, 5, SIAT1 ^p^	-
2007	A/Uruguay/716/07e	Ur07e	SPFCK1, Egg5	2008, 2009
2009	A/Perth/16/09	Pe09	MDCKX, 5	-
2009	A/Perth/16/09e	Pe09e	Egg6	2010, 2011, 2012
2011	A/Victoria/361/11	Vi11	MDCK2, SIAT1 ^p^	-
2011	A/Victoria/361/11e	Vi11e	Egg6	2013
2012	A/Texas/50/12	Tx12	C2, MDCK6, SIAT1	-
2012	A/Texas/50/12e	Tx12e	Egg5, Egg2	2014
2013	A/Switzerland/9715293/13	Sw13	SIAT, SIAT8	-
2013	A/Switzerland/9715293/13e	Sw13e	Egg6	2015
2014	A/Michigan/15/14	Mi14	MDCK1, SIAT6	-
2014	A/New Caledonia/104/14	NC14	MDCK1, SIAT4 ^p^	-
2014	A/Hong Kong/4801/14e	HK14e	Egg7	2016
2016	A/Newcastle/30/16	Nc16	SIAT1, SIAT4	-
2017	A/Kansas/14/17	Ka17	SIAT3, SIAT1	-
2018	A/Brisbane/60/18	Br18	SIAT3	-

a: The suffix e is used to indicate viruses that were grown in eggs. b: Passage cell type followed by number of passages where C = undefined cell line; MDCK = Madin Darby Canine Kidney cell line; SIAT = human 2,6-sialtransferase transfected MDCK cells; P = undefined passage; SPFCK = chicken kidney cell; X = unknown; superscript p = plaque selected. c: Year in which a strain was included in the Southern Hemisphere Influenza Vaccine.

**Table 2 viruses-14-00470-t002:** Age and sex of participants in each prior exposure group.

	Prior A(H3N2) Exposure Group			
	No Recent Exposure	Prior Infection	1–2 Prior Vaccinations	3–5 Prior Vaccinations
Study	Ha Nam	Ha Nam	HCW	HCW
N	22	64	13	28
Sex, F:M (%F)	14:8 (64)	41:23 (64)	9:4 (69)	18:10 (64)
Age Y, median (range)	48 (23–64)	48 (20–63)	37 (24–56)	41 (24–65)
Sample Day, median (range)				
Post-vaccination	21	21	21 (21–27)	22 (20–28)
Post-season	282 (280–282)	282 (280–282)	223 (210–225)	224 (210–227)

**Table 3 viruses-14-00470-t003:** Antigenic site variation among A(H3N2) strains in vaccines from 2011 to 2016.

		Amino Acids Substituted at Least Once by Antigenic Site and Position ^a^																			
Vaccine		Site A					Site B					Site C				Site D					E
Year	Strain	138	140	142	144	145	128	159	186	194	198	45	48	278	311	3	96	212	214	219	62
2016	HK14e	A	I	R	S	S	T+	Y	G	P	S	N	I	K	H	I	S	A	I	S	E
2015	Sw13e	S	R	G	N+	S	A	S	V	L	S	N	I	K	Q	L	N	A	I	Y	E
2014	Tx12e	A	I	R	N+	N	N	F	V	L	P	N	I	K	Q	L	N	A	I	F	E
2013	Vi11e	A	I	R	N+	N	T+	F	V	L	S	N	I	N	Q	L	N	A	I	X	E
2011/12	Pe09e	A	I	R	K−	N	T+	F	G	L	A	S	T	N	Q	L	N	T	S	S	K

a: Amino acids that differ from the prior vaccine strain are shaded yellow, while those that differ from the prevailing 2016 vaccine strain are colored red; + indicates introduction of a potential glycosylation site.

## Data Availability

HI titre data of participants who have consented to use of samples for future research will be made available on request and will be publically available for research that adheres to the original ethics approval conditions at https://melbourne.figshare.com/ (accessed on 22 August 2021) within one year of this publication. HA (+/− NA) sequences of influenza viruses included in the analyses are available on GISAID.

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
