# Peer review of "Opposing Effects of Prior Infection versus Prior Vaccination on Vaccine Immunogenicity against Influenza A(H3N2) Viruses"

_viruses, 2022, doi:10.3390/v14030470_

Round 1

Reviewer 1 Report

The study is well described and results clearly presented. 

According to the Introduction and Materials&Methods parts the study concerns A(H3N2) influenza virus, but the title suggests that "opposite effects of prior infection versus prior vaccination" are the same for influenza virus in general.  Consider to change the title for more adequate to the context.

Lines 87-91: Does mathematical modelling show that both the small and large antigenic distance attenuate protection? Check the meaning, the sentence is not clear. 

Table 1. The number of vaccine year in the line "2012 A/Texas/50/12e" is not complete.

Figures 2 and 3. Parts of the figure are signed with capital letters and in the text with lower cases.

Description of Figure 3 (line 260) - it should be: "Numbers per group are given in Table 2."

Limitations should be written straight on.

It seems that two studied groups were ethnically different. Doesn't it matter?

Reference no.14 - there are no details of publication.

Author Response

Consider to change the title for more adequate to the context. The title has been changed to specify that the study investigates A(H3N2) viruses.

Lines 87-91: Does mathematical modelling show that both the small and large antigenic distance attenuate protection? Check the meaning, the sentence is not clear. As stated in the sentence, this depends on whether considering antigenic distance between successive vaccine strains (where smaller antigenic distances are potentially problematic), or between vaccine strains and subsequent epidemic strains (where larger antigenic distances are problematic). We have added several words to the sentence to make this clearer as follows "This hypothesis has been supported by mathematical modelling, which predicts that a prior vaccine will negatively interfere with a current vaccine when the antigenic distance between the successive vaccine strains is small, and that this will attenuate protection when the antigenic distance between vaccine and subsequent epidemic strains is large [38, 40]"

The A/Texas/50/12 vaccine year has been completed, i.e. 2014.

The legends for Figures 2 and 3 has been corrected to use capitals for time-points as in the figures and to refer to Table 2 instead of Table 3.

Limitations should be written straight on. 

It seems that two studied groups were ethnically different. Doesn't it matter?

To address these two comments we have re-written the paragraph on limitations (lines 363-372) as follows: "The study has several limitations. Sample sizes were small, particularly of HCWs who had less than five prior vaccinations. The effects of prior infection versus vaccination reported here could be biased by ethnic and socioeconomic differences between community members from Viet Nam and HCWs from Australia. However, we could detect effects of vaccination history within the HCW cohort, consistent with our previous studies [25], and with studies showing effects on VE in various populations and countries [17, 18, 54]. HCWs infection histories were unknown,..."

Reference no.14 - there are no details of publication.  We have submitted a revision of this manuscript to Nat Med and will update the reference if accepted.

Reviewer 2 Report

In general, this is an extensive study on an interesting and yet controversial topic. The overall quality is high, although I am not sure if it is a good idea to combine two completely different studies into one and draw conclusions from that. My other major concern is regarding the practical implications of the results.

Specific comments:

Lines 23-24: the authors mention vaccine effectiveness in the rationale, and immunogenicity in the goals, which are two different things and cannot be interchanged. 

Lines 86-104: the introduction is extensive. This paragraph would be better suited in the discussion. 

Lines 120: please provide a more detailed description of the vaccines. Were they split, subunit or whole virion? This could impact vaccine efficacy. 

Line 174: provide reference for the assay. 

Line 175: why was the WHO assay modified?

Line 218: "differed somewhat" is not a scientific term. Please provide p values where appropriate in the text also. 

Line 313: again, immunogenicity and vaccine effectiveness are two different things and cannot be compared directly. 

Line 321: more effective than what? 

Discussion: The authors do not differentiate between the different inactivated vaccines, such as split, subunit or whole virion, with or without adjuvants. This is of major importance. I also do not see the practical implications. Suggesting to skip vaccination has no supporting evidence. In fact, vaccinating yearly against seasonal influenza has been repeatedly shown to be of enormous benefit. 

Author Response

We thank the reviewer for their care in identifying omissions and inconsistencies. We respond to each point below.

Lines 23-24: previous studies have demonstrated that both VE and immunogenicity can be attenuated by prior vaccination. We have amended line 23, it now reads "Prior vaccination can alternately enhance or attenuate influenza vaccine immunogenicity and effectiveness."

Lines 86-104: the introduction is extensive. This paragraph would be better suited in the discussion. We introduce the concept that effects of prior vaccination vary because it forms part of the rationale for comparing effects of prior vaccination versus prior infection. We introduce the concept of antibody focusing because it forms part of the rationale for investigations conducted in Section 3.3. We have moved sentences describing studies that provide evidence of antibody focusing to the discussion, We have also condensed several of the remaining sentences in paragraph 4 of the the Introduction. It is now shorter and reads: "Effects of prior vaccination on VE have varied between studies [27] and seasons [38, 39]. The antigenic distance hypothesis, supported by mathematical modelling, predicts that a prior vaccine will negatively interfere with a current vaccine when the antigenic distance between successive vaccine strains is small, and that this will attenuate protection when the antigenic distance between vaccine and subsequent epidemic strains is large [38, 40]. The antibody focusing hypothesis suggests that recalled memory B cells competitively dominate and focus responses on epitopes shared with previous strains, which could attenuate protection if epidemic strains acquire mutations within epitopes upon which antibodies are focused [44]. Monto et al. suggest that the term “negative antigenic interaction” best captures the immune mechanism underlying repeated vaccination effects, and stress the need to identify mechanisms that account for negative effects of prior vaccination versus potentially positive protective effects of prior infection [45]."

Lines 120: please provide a more detailed description of the vaccines. Were they split, subunit or whole virion? This could impact vaccine efficacy. These details have been included in lines 119-120 and 128-130 as follows: "...received southern hemisphere trivalent inactivated split egg-grown influenza vaccine (Vaxigrip, Sanofi)"; "...receive the 2016 southern hemisphere quadrivalent inactivated split egg-grown influenza vaccine (Fluarix Tetra 2016, Glaxo Smith Kline)"

Both vaccines are composed of inactivated split egg grown viruses, and comparative studies indicate that trivalent and quadrivalent (2 influenza B lineages) induce equivalent antibody responses against influenza A viruses [PMCID: PMC3750613; PMCID: PMC6605850; PMID: 29709447; PMID: 24022123]

Line 174: provide reference for the assay. This has been included = reference 46.

Line 175: why was the WHO assay modified?  We followed the protocols of our centre (WHO Collaborating Centre for Reference and Research on Influenza, Melbourne). Our centre uses 25 uL of sera, 25 ul virus and  25 ul RBC at 1% whereas the WHO Reference Manual specifies 25 uL of sera, 25 ul virus and 50ul of 0.75% RBC. The final percentage of (RBC 0.375% and 0.333%, respectively) is very similar.

Line 218: "differed somewhat" is not a scientific term. Please provide p values where appropriate in the text also. The landscapes are estimated models of titre fit across stains, and we provide confidence intervals for the model, but cannot provide single p values. We have removed the first sentence of Section 3.2 and revised the second sentence as follows: "Compared to participants with no recent exposure, participants with prior vaccination had higher pre-vaccination antibody titres against 2009-2014 strains, approximating strains present in prior vaccines (Figure 2a, b, Figure S1a)."

Line 313: again, immunogenicity and vaccine effectiveness are two different things and cannot be compared directly. We acknowledge that these are different things with different assessment measures. However, it is established that immunogenicity and effectiveness are associated [PMCID:PMC2130285; PMCID: PMC2851702; PMID: 32484513], and therefore feel that it is valid to discuss similarities in the ways that they are linked to exposure history.

Line 321: more effective than what? We have added "than standard egg-grown virus vaccines"

Discussion: The authors do not differentiate between the different inactivated vaccines, such as split, subunit or whole virion, with or without adjuvants. This is of major importance. I also do not see the practical implications. Suggesting to skip vaccination has no supporting evidence. In fact, vaccinating yearly against seasonal influenza has been repeatedly shown to be of enormous benefitWe have added the clarification that this study assesses inactivated split egg-grown vaccine to the first paragraph of the Discussion. We had suggested that it will be important to consider how prior exposure impact the immunogenicity and  effectiveness of cell-grown and recombinant vaccines (line 318).  We have adjusted lines 315 - 318 to include adjuvanted vaccines in the list. "Inactivated cell-grown, recombinant HA, and adjuvanted vaccines have been developed in recent years, and early evaluations indicate that they are more immunogenic [53-57] and effective [58-60] than standard egg-grown virus vaccines. "

We fully appreciate the reveiwers concerns about the topic of deviating from annual vaccination. We did not intend to suggest that the annual vaccination policy should be changed. However, we do think that there is a case for research that investigate alternate regimens.  One of the authors on this Manuscript (Sheena Sullivan) has just completed a systematic review and meta analysis on the effects of repeated vaccination on vaccine effectiveness for the WHO Strategic Advisory Group of Experts (SAGE Working Group on Influenza Vaccines). This will be available via the Weekly Epidemiological Record on 17.12.2021. Estimated VE against A(H3N2) was lower in people vaccinated in the current and prior year (20%, 12-27%) compared to those vaccinated in current year only (37%, 29-45%), but higher that in those vaccinated in prior year only (8%, -4-8%). This goes against a strategy of vaccinating in alternate years. However, the data presented here, indicates that immunogenicity is particularly poor in people vaccinated 3 to 5 years in a row as opposed to the prior 1 or 2 years only. It will therefore be important to determine VE among this group compared to current or prior year only groups. We have ammended the last sentences of paragraph 1 of the Discussion as follows "While vaccine effectiveness against A(H3N2) viruses is poor and attenuated by prior vaccination, current estimates suggest that vaccination in the current and prior season affords better protection against A(H3N2) than being vaccinated in the prior season only [61]. This suggests that it is better to vaccinate annually than in alternate years. However, there have been no formal comparisons of VE among people vaccinated in alternate versus successive years, to determine whether there is any benefit to protection in the vaccinated years, and whether this outweighs the increased risk of infection in unvaccinated years. Additionally, the results presented here indicate that vaccine immunogenicity was particularly poor in people who had been vaccinated during three to five as opposed to one to two prior years, indicating a need to determine the impact of multiple years of vaccination on VE."

Reviewer 3 Report

The manuscript “ Opposite effects of prior infection versus prior vaccination on  influenza vaccine immunogenicity” has been reviewed.

Influenza represents a well-known but still intriguing public health problem with an available yet not total satisfactory preventive vaccine.  Population groups at risk of severe complications or death are recommended to receive yearly vaccination. A series of factors influence yearly observed IVE estimates besides discordant virus antigenic characteristics. Apart from the antigenic match with the vaccine circulating strain, recent studies have suggested that the protective effect of the current season ́s influenza vaccine could be also influenced by previous vaccinations as well as by previous infection through B cell mediated memory recall.

This is an extremely interesting target issue to study in order to achieve better understanding of the immune response in the real world and to, hopefully, be able to attain a universal vaccine. Aware of the complexity of the study, there are some queries I would like to have clarified:

  1. The fact that strains from 1968 to 2018 (35 strains) are used is because those are the ones available ? May be this fact could be included in the introduction or in the methods  3 section.
  2. Methods section : Participant inclusion and exclusion criteria
    1. Could you lease clarify why 2 participants infected with AH3N2 in 2008 were excluded from the prior infection group?
    2. What does A (H3N2) + ILI stand for ?
    3. The final HCW selected box should be 28 3-5 prior vaccinations, so it would include 8 3-4 vaccinations and the random selection of 20 with 5 prior vaccinations. If this is so, a new box would be more clear.

Minor comments :

Introduction

pg 2 line 87 : after….. distances…… there is a 41 in superscript. If this is a reference should be in brackets yet it preceeds reference 40 which is below. Please check and correct.

Materials and methods

pg 4 line 152 : a A (H3N2) + ILI , is unclear, please define the term before using it :

Cases presenting Influenza like Illness after vaccination, caused by confirmed A(H3N2) influenza virus  [A (H3N2) + ILI] were excluded from both comparison groups…….

Line 164 write out first mention of abbreviation MDCK-SIAT cells

Line 174 : Include reference for WHO Global Influenza Surveillance Network protocol

Results

Pg 8 Line 252 : A reference to the ascertainment of differences between egg and cell grown vaccines is needed

Pg 10 Line 286  after epitopes the reference number needs brackets  [44]

References

The working groups’ names are to be corrected , for example in

 number 34 it should read as:

,US Influenza Vaccine Effectiveness (Flu VE) Network, the Influenza Hospitalization Surveillance Network, and the Assessment Branch, Immunization Services Division, Centers for Disease Control and Prevention

Or in number 30

, Primary Health Care Sentinel Network of Navarre , Network for Influenza Surveillance in Hospitals of Navarre

Or number 39

, and I‐MOVE primary care multicentre case‐control team

Author Response

  1. The fact that strains from 1968 to 2018 (35 strains) are used is because those are the ones available ? May be this fact could be included in the introduction or in the methods  3 section. We have included this information in the Methods, lines 178-180 as follows: "Viruses were selected to represent each main antigenic or genetic cluster detected between 1968, when A(H3N2) emerged in humans, and 2018, four years after the strain included in the 2016 vaccine (A/Hong Kong/4801/2014). "
  1. Methods section : Participant inclusion and exclusion criteria
    1. Could you please clarify why 2 participants infected with AH3N2 in 2008 were excluded from the prior infection group? We wanted to compare participants that had been exposed to a similar range of strains via infection versus vaccination. Health Care Workers had 5-year vaccination histories, since 2011 when the vaccine strain was A/Perth/16/2009. We excluded the two people from Ha Nam who were last infected in 2008 with strains that circulated prior to A?Perth/16/09, namely A/Brisbane/10/2007-like strains.  We have amended lines 149-152 as follows "Ha Nam vaccinees were excluded if aged > 65 Y. We further excluded two participants who were last infected in 2008 with an A/Brisbane/10/2007-like (H3N2) virus in order to match the range of strains that HCWs were exposed to by vaccination between 2011 and 2015"
    2. What does A (H3N2) + ILI stand for ? ILI with A(H3N2) virus infection confirmed by RT PCR. This has now been specified in the Methods, line 147.
    3. The final HCW selected box should be 28 3-5 prior vaccinations, so it would include 8 3-4 vaccinations and the random selection of 20 with 5 prior vaccinations. If this is so, a new box would be more clear. Figure 1 has been ammended as suggested.

pg 2 line 87 : after….. distances…… there is a 41 in superscript. If this is a reference should be in brackets yet it preceeds reference 40 which is below. Please check and correct. This has been corrected.

pg 4 line 152 : a A (H3N2) + ILI , is unclear, please define the term before using it. This has now been specified, now line 147.

Line 164 write out first mention of abbreviation MDCK-SIAT cells. This has been specified here and in the footnotes to Table 1.

Line 174 : Include reference for WHO Global Influenza Surveillance Network protocol. This has been included.

References: The working groups’ names are to be corrected. These have been corrected.

Round 2

Reviewer 2 Report

I maintain my main concern that the authors pooled two different studies together and their conclusions are drawn from that. Also, the conclusions are not supported by the findings. 

I also don't understand their discussion on "egg grown virus vaccines": "Inactivated cell-grown, recombinant HA, and adjuvanted vaccines have been developed in recent years, and early evaluations indicate that they are more immunogenic [53-57] and effective [58-60] than standard egg-grown virus vaccines. "

Contrary to what the authors state, there are adjuvanted vaccines that are "egg-grown", such as in this paper:  Baseline CD3+CD56+ (NKT-like) Cells and the Outcome of Influenza Vaccination in Children Undergoing Chemotherapy (nih.gov) 

The EMA actually revised its guidelines for influenza vaccine approvals to focus on efficacy, not immunogenicity, precisely because they correlate poorly. For inactivated influenza vaccines containing viral HA, an HI titre of 1:40 was previously suggested to represent a reasonable statistical correlate for an efficacy of 50-70% against clinical symptoms of influenza based on challenge studies in healthy adults. Since then, evidence has emerged to indicate that there remains a need to better define correlates of protection against influenza, which potentially may vary according to individual characteristics, populations, specific age groups (e.g. the paediatric 
population) and vaccine types.